# Comparative Study of the Pharmacological Properties and Biological Effects of *Polygonum aviculare L. herba* Extract-Entrapped Liposomes versus Quercetin-Entrapped Liposomes on Doxorubicin-Induced Toxicity on HUVECs

**DOI:** 10.3390/pharmaceutics13091418

**Published:** 2021-09-07

**Authors:** Mariana Mureşan, Diana Olteanu, Gabriela Adriana Filip, Simona Clichici, Ioana Baldea, Tunde Jurca, Annamaria Pallag, Eleonora Marian, Adina Frum, Felicia Gabriela Gligor, Paula Svera, Bogdan Stancu, Laura Vicaș

**Affiliations:** 1Department of Preclinical Disciplines, Faculty of Medicine and Pharmacy, University of Oradea, 10 Piata 1 Decembrie Street, 410073 Oradea, Romania; marianamur2002@yahoo.com; 2Department of Physiology, “Iuliu Hatieganu” University of Medicine and Pharmacy, 1–3 Clinicilor Street, 400006 Cluj-Napoca, Romania; ariana_di@yahoo.com (D.O.); simonaclichici@yahoo.com (S.C.); baldeaioana@gmail.com (I.B.); 3Department of Pharmacy, Faculty of Medicine and Pharmacy, University of Oradea, 29 Nicolae Jiga Street, 410028 Oradea, Romania; jurcatunde@yahoo.com (T.J.); annamariapallag@gmail.com (A.P.); marian_eleonora@yahoo.com (E.M.); laura.vicas@gmail.com (L.V.); 4Faculty of Medicine, Lucian Blaga University Sibiu, Lucian Blaga Street, No. 2A, 550169 Sibiu, Romania; adinafrum@gmail.com (A.F.); feligligor@yahoo.fr (F.G.G.); 5INCEMC-National Institute for Research and Development in Electrochemistry and Condensed Matter-Timisoara, No. 144 Dr. A. Paunescu Podeanu Street, 300569 Timisoara, Romania; paulasvera@gmail.com; 62nd Department of General Surgery, Iuliu Hatieganu University of Medicine and Pharmacy, 400006 Cluj-Napoca, Romania; bstancu7@yahoo.com

**Keywords:** liposomes, quercetin, *Polygonum aviculare* L. extract, doxorubicin, endothelial cells, cell death

## Abstract

This study aimed to evaluate the comparative biological effects of *Polygonum aviculare L. herba* (PAH) extract and quercetin-entrapped liposomes on doxorubicin (Doxo)-induced toxicity in HUVECs. HUVECs were treated with two formulations of liposomes loaded with PAH extract (L5 and L6) and two formulations of liposomes loaded with quercetin (L3 prepared with phosphatidylcholine and L4 prepared with phosphatidylserine). The results obtained with atomic force microscopy, zeta potential and entrapment liposome efficiency confirmed the interactions of the liposomes with PAH or free quercetin and a controlled release of flavonoids entrapped in all the liposomes. Doxo decreased the cell viability and induced oxidative stress, inflammation, DNA lesions and apoptosis in parallel with the activation of Nrf2 and NF-kB. Free quercetin, L3 and L4 inhibited the oxidative stress and inflammation and reduced apoptosis, particularly L3. Additionally, these compounds diminished the Nrf2 and NF-kB expressions and DNA lesions, principally L4. PAH extract, L5 and L6 exerted antioxidant and anti-inflammatory activities, reduced γH2AX formation and inhibited extrinsic apoptosis and transcription factors activation but to a lesser extent. The loading of quercetin in liposomes increased the cell viability and exerted better endothelial protection compared to free quercetin, especially L3. The liposomes with PAH extract had moderate efficiency, mainly due to the antioxidant and anti-inflammatory effects and the inhibition of extrinsic apoptosis.

## 1. Introduction

The cellular mimetic function of liposomes has attracted the attention of researchers in the medical and pharmaceutical fields due to their ability to be personalised and their properties to counteract the major disadvantages of bioactive molecules with regards to decreased stability, limited membrane permeability, short half-life and low bioavailability. Liposomes are closed vesicular structures composed of one or more phospholipid layers, which are formed when membrane lipids such as phosphatidylcholine (PC) and cholesterol (CHO) are dispersed in an excess of water. Liposomes have the ability to be selectively absorbed by tissues rich in the reticuloendothelial system, including the liver, spleen and small bones. This property underpins the mechanism of vectorisation of the bioactive substance and its transport in the systemic circulation. The optimisation of drugs through changes in their bioactive compounds by way of entrapment in liposomal particulate systems brings benefits to pharmacotherapy by capitalising on their pharmacological potential. Therefore, the aims of the study are to show improvements in intracellular penetration and cell availability and an increase in specificity of action for bioactive substances entrapped in liposomes with the lowest possible size distribution. The method proposed is an ultrasound-assisted thin-film hydration method. Numerous compounds, such as polyphenols or anthocyanins, are secondary metabolites generated by plants as a defence mechanism against infections or rough pedoclimatic conditions. These phytochemicals are bioactive agents with proven antioxidant and anti-inflammatory properties and proapoptotic, antimicrobial and antifungal effects, which can reduce the toxicity of free radicals in the human body and preserve human health [1,2,3].

The polyphenols’ role in the modulation of the metabolic, functional and toxicity features in the human body are limited by low oral bioavailability—specifically, absorption and metabolism—in relationship with the chemical structure, molecular size, degree of polymerisation and water solubility [4]. These limits are exceeded by the incorporation of polyphenols into different drug delivery systems, including cyclodextrin [5], simple emulsions, gels, solid lipid nanoparticles, lipid nanocapsules, nanoemulsion [6] or liposomes [7,8].

The vegetal product used in this study originates from the *Polygonum aviculare* L. species, also known as common knotgrass. The *Polygonum aviculare* L. species is part of the Polygonaceae family [9,10,11,12,13], and their active compounds are identified especially in the aerial part of the plant. *Polygonum aviculare L. herba* (PAH) was harvested during the blooming season in June. Numerous studies have demonstrated the anti-inflammatory [14,15], antimicrobial and antifungal effects of the plant [14,16] due to flavonoids, quinones, anthraquinones phenylpropanoids and terpenoids from its composition. Their antioxidant and protective effects may be useful properties in counteracting the toxic action on normal cells of different drugs, including antibiotic and chemotherapeutic agents.

Doxorubicin (Doxo) is a commonly anthracycline chemotherapeutic drug used for a long time in the treatment schemes of malignant tumours, such as bladder, colon and breast, or in Kaposi’s sarcoma, lymphoma and leukaemia [17,18]. Although it is very effective, it has a major disadvantage due to cardiotoxicity and endothelial vascular toxicity [19], which often result in heart failure and increase the noncancerous mortality [20]. The mechanisms of toxicity are owed to intracellular and mitochondrial redox imbalance [21] and intrinsic apoptotic induction by the activation of caspase-3 and p53, as well as extrinsic pathway initiation via the Fas receptor and ligand stimulation. Doxo also inhibits the nuclear factor related to erythroid factor 2 (Nrf2), a transcription factor that controls the antioxidant genes and cytoprotective enzymes expressed in the cardiovascular system [22,23,24], reduces nitric oxide synthesis and upregulates nuclear factor (NF)-kB [25]. Therefore, the therapeutic strategy that associates Doxo with a natural antioxidant may be a good option to diminish the cardio and vascular endothelial toxicities [26,27].

Given these data, this study aims to optimise the preparation of bioactive substances entrapped in liposomes with the lowest possible size distribution and highest penetrability for biomedical applications. After the characterisation of liposomes loaded with quercetin and the PAH extract, their efficacy in the inhibition of Doxo-induced toxicity was tested comparative only with quercetin and the PAH extract on human endothelial cells (HUVECs). The mechanisms involved in the endotheliotoxicity of Doxo was evaluated by oxidative stress and inflammation markers, as well as caspases-3, -8 and -9; DNA lesions and transcription factor expressions. Quercetin was chosen for comparison, because it is an important component of the tested extract, and because it has demonstrated endothelial-protective properties on doxorubicin-induced toxicity [28]. Thus, on isolated aortic rings incubated with Doxo, quercetin restored the normal vascular contraction and relaxation impaired by Doxo exposure and diminished the ROS generation. In addition, quercetin inhibited the ERK/MAP-kinase activation and, thus, diminished the ROS-induced cardiomyopathy [29].

## 2. Materials and Methods

### 2.1. Reagents

The liposomes loaded with polyphenols from the *Polygonum aviculare* L. herb extract were prepared by the hydration method by using phosphatidylcholine, phosphatidylserine and cholesterol purchased from Sigma-Aldrich (Milan, Italy). The *Polygonum aviculare* L. herb was harvested from the Crișana region, Bihor County. A specimen from this species was kept in the herbarium of the Faculty of Medicine and Pharmacy Oradea, registered in the NYBG Steere Herbarium, code UOP 05168. Ethanol, n-butanol and phosphate buffer were obtained from Farmachim 10 SRL Ploiesti (Bucharest, Romania), while methanol pro-analysis (Promochem), chloroform pro-analysis, Folin-Ciocâlteu’phenol reagent and 2-thiobarbituric acid EDTANa2 were purchased from Merck KGaA (Darmstadt, Germany). Trolox was sourced from Chemical Industry Co Ltd. (Tokyo, Japan) and 2,2-diphenyl-1-picrylhydrazyl; 2,2′-azino bis (3-ethylbenzothiazoline-6-sulfonicacid) diammonium salt; neocuproine and the following standards: 7-methoxicumarin, gallic acid, chlorogenic acid, caffeic acid, ferrulic acid, rosmarinic acid, ellagic acid, trans-p-coumaric acid, bergapten, delphinidin 3-rutinchlorid, diosmin, hyperoside, isopimpinellin, luteolin-7-glucoside, myricetin, quercetin, rutin and t-resveratrol were from Sigma Aldrich. Antibodies against NF-kB p65 (Ser536) (93H1), phospho (p)NF-kB, Nrf2 and glyceraldehyde 3-phosphate dehydrogenase (GAPDH) and the secondary peroxidase-linked antibodies were from Santa Cruz Biotechnology (Delaware Ave, Santa Cruz, CA, USA), while phosphorylated histone H2AX (pS139) (γH2AX) was from Stressgen Bioreagents Corporation (Victoria, BC, Canada). ELISA tests for IL-6 and caspases-3, -8 and -9 were purchased from Elabscience (Houston, TX, USA), and the Bradford total protein concentration assay was from Bio-Rad (Hercules, CA, USA). The compounds used were of adequate purity, attested by analysis bulletins issued by the manufacturer.

### 2.2. Obtaining the Lyophilised Extract of Polygonum aviculare L. herba

In order to obtain the fluid extract, the maceration extraction method at a temperature of 20 °C was used. PAH was dried pre-procedure in a dry room, sheltered from sunlight and with an average temperature of 22 °C. The vegetal product was fragmented and sieved through a no. III pharmaceutical sieve [30,31]. The solvent used for extraction was a mix of ethyl-alcohol and distilled water 30% (*v/v*). The mass ratio of the vegetal product to the solvent was 1:10 (*m*/*m*). The ethyl-alcohol was removed by evaporation on a rotavapour model, Hei-VAP Precision–Platinum 3, at a temperature of 40 °C, 80 rpm and 200 mBars. The aqueous fraction left behind was frozen at a temperature of −80 °C and dried using the ALPHA 2-4 LSC plus laboratory lyophiliser.

### 2.3. The Phytochemical Investigation of the Lyophilised PAH Extract

The physicochemical characterisation by the HPLC-PDA method

The identification and quantification of the bioactive phenolic compounds from the PAH extract was performed by using a Shimadzu Nexera-i LC–2040C 3D plus liquid chromatograph system equipped with a photodiode array detector (PDA). A Phenomenex C18 (2) 100 A, 150 mm × 4.6 mm × 5 µm column was used, and it was kept at 30 °C. The mobile phases used for elution consisted of methanol (A) and formic acid 0.1% (B). The gradient program used was: 5% A and 95% B from 0 to 3 min, 25% A and 75% B from 3 to 6 min, 37% A and 63% B from 9 to 13 min, 54% A and 46% B from 18 to 22 min, 95% A and 5% B from 26 to 29 min and 5% A and 95% B from 30 to 36 min. The flow rate was 0.5 mL/min, and the injection volume was 10 μL. The detection was performed at multiple wavelengths: 254, 270, 275, 326, 337 and 360 nm. The polyphenols from the extract were identified by comparing the retention times from the extract chromatograms with the ones from the standard solution chromatograms.

The contents of the bioactive compounds

The contents of the bioactive compounds of the PAH extract (respectively, the total polyphenol and total flavonoid contents) were assessed. In order to evaluate the total polyphenol content, the Folin–Ciocâlteu method was used [32,33]. This method is based on the electron transfer reaction, measuring the reductive capacity of an antioxidant. The results of the Folin–Ciocâlteu method were very well-correlated with the results obtained from other antioxidant analyses, such as ABTS and DPPH. The total polyphenol content was calculated as the gallic acid equivalent (GAE/100 g) of dried plant based on the calibration line of gallic acid (5–500 mg/L, Y = 0.0027x − 0.0055, *R*^2^ = 0.9999). All determinations were performed in triplicate.

For the evaluation of the total flavonoid content, the aluminium chloride colorimetric method was used [34,35,36,37]. Quercetin was used for the standard calibration curve. The stock quercetin solution was prepared by dissolving 5-mg quercetin in 1-mL methanol, followed by a standard quercetin solution preparation with serial dilutions using methanol (5–200 µg/mL). The absorbance was read at a wavelength of 420 nm with an UV-Vis Varia spectrophotometer. The total flavonoid concentration was calculated based on the calibration line (*Y* = 0.0162x + 0.0044, *R*^2^ = 0.999) and expressed as the quercetin equivalent (QE) mg/100 g in dried plant. All determinations were performed in triplicate.

### 2.4. Evaluation of Antioxidant Activity

The antioxidant activity of the PAH extract using the methods: 2,2-diphenyl-1-picrylhydrazyl (DPPH), ABTS (2,29-azinobis-(3-ethylbenzothiazoline-6-sulfonic acid–ABTS•1), CUPRAC (Cupric-Reducing Antioxidant Capacity) and FRAP (Ferric Reducing Antioxidant Power) [38].

### 2.5. Preparation of Liposomes

Six liposome formulations were prepared in the following way: 2 formulations of un-loaded liposomes (L1 and L2), 2 formulations of liposomes loaded with quercetin (L3 and L4) and 2 formulations of liposomes loaded with PAH extract (Table 1). The following lipids to prepare the liposomes were used: phosphatidylcholine (PC), phosphatidylserine (PS) and sodium cholate (CoN). The method chosen for preparation was the thin-film method [39]. In order to obtain comparable results between the liposomes loaded with active substances, the liposomes were loaded with 10-mg quercetin (L3 and L4), as well as 100-mg PAH extract (0.676-µg QE). As a first step, 5 mL of organic solvent was dissolved and mixed in chloroform, thus assuring adequate homogeneity and resulting in a clear lipid solution. In the second step, quercetin and PAH extract were included for entrapment in the lipid solution. After the compounds dissolved in chloroform, the solvent was removed, resulting in the formation of a lipid film in each receptacle. The organic solvent was removed by evaporation at a temperature of 40 °C in a rotavapour model, Hei-VAP Precision–Platinum 3.80 rpm and 200 mBars. The hydration of the dry lipid film was achieved by adding 2 mL of phosphate buffer solution with a pH = 7.4, followed by centrifugal shaking/stirring in a Hettich Model Universal-320R centrifuge at 10,000 rpm for 30 min and, finally, ultrasonication for another 30 min at room temperature. This process resulted in nanometric structures after ultrasonication.

### 2.6. Characterisation of Liposomes

Dynamic light scattering (DLS) and zeta potential (ZP) measurements were done using a Zetasizer Nano ZS (Malvern Instruments, Worcestershire, UK). Polystyrene cells with 1-cm optical path were used for size measurements. This technique was used to determine the particle size distribution profile, the shape of the particles being approximated as spherical. The advantage of this method is that it covers a wide range of dimensions (10–3–10–9 m). This method does not directly measure the particle size, it is calculated based on their effect on light diffusion, the results being conclusive in the case of spherical particles. For this reason, some uncertainties may arise about the sizes of the particles if they deviate from a spherical shape [40]. Atomic Force Microscopy (AFM) images were obtained using a Scanning Probe Microscopy Platform (MultiView-2000 system, Nanonics Imaging Ltd., Jerusalem, Israel) in normal conditions (24–25 °C). During the analysis, it was used tip-dopped with chromium, with a 20-nm radius and 30-40-KHz resonance. Before the analysis, all samples (L1–L6) were exposed to ultrasonication for one hour, followed by drop-casting on an AFM glass support (purchased from Nanonics Imaging Ltd., Jerusalem, Israel). The samples were dropped on the support twice, followed by a drying process at room temperature.

### 2.7. Liposome Entrapment Efficiency

In order to know the degree of liposomal quercetin entrapment, as well as the total flavonoids from the PAH extract, the liposomes were centrifuged at 10,000 rpm for 30 min to separate the substances. Following this, 0.5 mL of Triton X-100 (0.5% *v/v*) was added to each specimen to decompose the lipid membranes. Each specimen was diluted with methanol and then filtered. To evaluate the quantity of entrapped PAH extract, we determined the total flavonoid content using the colorimetric method described in this article in Section 3.2, using the same calibration curve. Blank probes were used as witness probes. The entrapment efficiency (*EE*%) was calculated with the equation below:*EE*(*%*) = *M*/*M_t_* × 100
where *M* is the quantity of flavonoids loaded into the liposomes, and *M_t_* is the quantity of flavonoids from the extract [37]. The entrapment efficiency (*EE*%) of the lipid vesicles loaded with quercetin and the PAH extract, respectively, as well as their stability over time, were performed at three different storage intervals—namely, at 10, 30 and 60 days.

### 2.8. In Vitro Studies on the Release of Flavonoids Entrapped in Liposomes Using a Franz Cell System

The in vitro diffusion experiments were conducted according to the FDA SUPAC-SS recommendations, using a diffusion system with 6 Franz diffusion cells (Microette-Hanson System, model 57-6AS9, Copley Scientific Ltd., Nottingham, UK), with a diffusion surface of 1.767 cm^2^ and a volume of 6.5 mL for the receptor chamber. The receptor chamber in each diffusion cell was filled with saline phosphate buffer (pH 7.4) mixed with 30% freshly prepared ethanol, which was warmed and deaerated [41].

The synthetic membranes were hydrated by immersion in the receptor medium for 30 min before use and then placed between the donor and acceptor compartments of the Franz diffusion cell. Approximately 0.300 g of the specimen were weighed in the dosing capsule of each diffusion cell, followed by application onto the superior part of the Franz diffusion cell membrane surface. The diffusion cells were well-sealed by fixing the dosing capsule with a clamp, preventing the evaporation of the vehicle and assuring the integrity of the formulation for the entire duration of the study. During the test, the system was maintained at 32 ± 1 °C, and the receptor medium was continually stirred (600 rpm) with the help of a magnetic stirrer to avoid the effects of the diffusion layer. The diffusion membranes are made of polysulfone with a diameter of 25 mm and a pore size of 0.45 μm (Tuffryn^®^, PALL Life Sciences HT-450, batch T72556). Before starting the study, the diffusion membranes were wetted by immersion in the receptor medium and kept in contact for 15 min. Next, 0.5 mL of the receptor solution were automatically withdrawn at 30 min and 45 min and 1, 3, 4, 5, 6, 8, 12 and 24 h and were replaced with a fresh receptor medium to maintain a constant volume of 6.5 mL during the test.

The specimens underwent a UV spectrophotometric analysis with a PG INSTRUMENTS model UV–VIS T 92+ spectrophotometer at 510 nm, which corresponded to the maximum absorption of quercetin in saline phosphate buffer (pH = 7.4) with 30% ethanol. The results were evaluated by using the calibration curve for quercetin. Each formulation was tested in triplicate and the data presented as the mean ± SD (standard deviation). The cumulative release rate or cumulative drug release (CDR) was calculated using the equation:CDR (%) = *Q_n_*/*Q_t_* × 100
where *Q_n_* is the quercetin content and total flavonoid content at time *n*, and *Q_t_* is the initial entrapped substance content in the lipid vesicles [39].

### 2.9. In Vitro Studies

Cell cultures

Commercial human umbilical vein endothelial cells (HUVEC) obtained from the European Collection of Cell Cultures (ECACC, Porton Down, Salisbury, UK) were used to test the toxicity of Doxo and the effects of free quercetin, PAH extract and liposomes prepared with quercetin or the PAH extract. The cells were multiplied in RPMI medium, supplemented with 10% foetal calf serum (FCS), antibiotics and anti-mycotics (Biochrom AG, Berlin, Germany) in a humidified CO_2_ incubator at 37 °C. The surfaces markers of the cells, ICAM-1, CD29, CD34, CD73, CD90 and CD105, were analysed using flow cytometry, as previously published [42]. Cell cultures in the 23rd to 26th passages were used. All reagents were purchased from Sigma Aldrich, Co (Heidelberg, Germany).

Viability assay and cell lysates

The cells’ survival under exposure to free quercetin; PAH extract and liposomes loaded with quercetin or PAH extract (L3, L4, L5 and L6) were tested through the colourimetric measurement of formazan using the CellTiter 96^®^ AQueous Non-Radioactive Cell Proliferation Assay (Promega Corporation, Madison, WI, USA). HUVECs were cultivated at a density of 10^4^/well in 96-well plaques (TPP, Trasadingen, Switzerland) for 24 h and then treated for 24 h with different doses of free quercetin (ranged between 0.001 and 100 µg/mL), PAH extract (ranged between 0.006 and 0.678 µg/mL), L3 and L4 (ranged between 0.001 and 100 µg/mL) and L5 and L6 (ranged between 0.006 and 0.678 µg/mL) formulations. For MTS, the optical density values were read at an absorbance of 540 nm using an ELISA plate reader (Tecan, Männedorf, Switzerland). Data are presented as a mean of OD540 ± SD. All the experiments were conducted in triplicate. Untreated cultures exposed to medium were used as controls.

For obtaining cell lysates, HUVECs cells seeded on Petri dishes at a density of 10^4^/cm^2^ were exposed for 24 h to quercetin; PAH extract and the L3, L4, L5 and L6 formulations at a dose of 0.01 µg/mL concomitant with 2-µg Doxo/mL. Untreated cells were used as controls. After 24 h, the cells were collected, and the cell lysates were obtained using the standard procedure [43]. Protein concentrations in the samples were evaluated by the Bradford method, according to the manufacturer’s specifications (Bio-Rad, Hercules, CA, USA). All the experiments were performed in triplicate.

Oxidative stress, apoptosis and inflammation markers assessment

For the evaluation of oxidative stress, malondialdehyde (MDA) was assessed using the fluorimetric method with 2-thiobarbituric acid [44]. The results were expressed as nmoles/mg protein. In order to evaluate the apoptosis, the caspases-3, -8 and -9 levels in the supernates were quantified by ELISA kits according to the manufacturer’s instructions. As a marker of inflammation, the IL-6 content in the supernates was also determined. The results were expressed as ng/mL.

DNA lesions and transcription factors expressions

The expression of transcription factors NF-kB and its phosphorylated form (pNF-kB), as well as Nrf2, were evaluated by Western blotting, as previously described [45]. For the quantification of DNA lesions, the expression of γH2AX was also evaluated by Western blotting. Lysates (20-mg protein/lane) were separated by electrophoresis on SDS PAGE gels and then transferred to polyvinylidenedifluoride membranes. Blots were blocked and then incubated with antibodies against NF-kB 65 (Ser536) (93H1), phosphorylated NF-kB (pNF-kB), Nrf2, γH2AX and GAPDH. Then, the blots were washed and incubated with the corresponding secondary peroxidase-linked antibodies. GAPDH was used as the protein loading control. Proteins were detected using the Supersignal West Femto Chemiluminiscent substrate (Thermo Fisher Scientific, Bath, UK) and a Gel Doc Imaging system equipped with an XRS camera and Quantity One analysis software (Bio-Rad, Hercules, CA, USA).

### 2.10. Statistical Analyses

The statistical analysis was conducted by using one-way ANOVA followed by Dunnett’s multiple range test. All reported data were expressed as the mean of triplicate measurements ± SD, and a *p*-value less than 0.05 was considered statistically significant.

## 3. Results and Discussion

### 3.1. The Physicochemical Characterisation of PAH Extract by the HPLC-PDA Method

The identification and quantification of bioactive phenolic compounds from the PAH extract was performed by using a Shimadzu Nexera-i LC–2040C 3D plus liquid chromatograph system equipped with a photodiode array detector (PDA). The polyphenols from the PAH extract were identified by comparing the retention times from the extract chromatograms with the ones from the standard solution chromatograms. The results obtained for the determination of the PAH extract content are presented in Table 2.

The most important compounds identified were phenolic acids, and the highest amounts of phytochemicals found in the PAH extract were: rosmarinic acid (2.65 ± 16.089 mg/100-mg DW), followed by delphinidin 3-rutinoside chloride (1.034 ± 6.326 mg/100-mg DW) and quercetin (0.603 ± 19.543 mg/100-mg DW). Ferulic acid, bergapten, diosmin, hyperoside, isopimpinellin, luteolin 7-glucoside and trans resveratrol were not detected in our extract.

The results obtained showed the differences from the concentrations of bioactive compounds previously identified by other authors [12,46,47,48]. Other studies found in *Polygonum aviculare* L. species large amounts of phenolic compounds [49], the total quantity of the flavonoids being between 0.1 and 1% and, rarely, up to 2.5–3%. The principal flavonoids identified were derivates of kaempferol, quercetin, myricetin and, particularly, avicularin (quercetin-3-O-arabinoside in a proportion of approximately 0.2%), juglanin (kaempferol-3-O-arabinoside), rutin, apigenin, hyperoside and quercetin-3-galactoside, as well as vitexin and isovitexin. The differences found may probably be due to the pedoclimatic conditions in the area of origin of the plant materials.

### 3.2. Bioactive Compound Content and Antioxidant Activity of PAH Extract

Although all the phenolic compounds were measured, not all these compounds present in nature give the same response. The quantification by HPLC–PDA was well-correlated with the total polyphenolic content. However, the Folin–Ciocâlteu method usually results in higher readings when compared to other methods. The total polyphenols determined by Folin–Ciocâlteu and expressed as mg GAE/100-g dry extract (DW) identified in the PAH extract were 19.08 ± 0.234, while the total flavonoids were 0.676 ± 0.753 mg QE/100-g DW. The antioxidant capacity through the DPPH, CUPRAC and ABTS methods had high values compared to the other plant extracts studied. Thus, lyophilised PAH extract showed the following values of antioxidant activity measured by different methods: DPPH: IC_50_ = 51.18 μg/mL, ABTS: 70.98 ± 1.20-mmol TE/g DW, FRAP: 236.57 ± 2.15-µmol TE/g DW and CUPRAC 448.92 ± 3.21-µmol TE/g DW. Hsu [11] found in *Polygonum aviculare* L. extract powder a total flavonoid content of 112.7 ± 13 µg/g, while the total polyphenolics content was 677.4 ± 62.7 µg/g. The antioxidant activities examined by different methods showed a value of IC_50_ equal to 50 µg/mL in the free radical scavenging assay, 0.8 µ/mL in the superoxide radical scavenging analyses and 15 µg/mL in the lipid peroxidation assays, respectively [11]. These results confirmed a high content of active compounds and a good antioxidant capacity of the PAH extract.

### 3.3. Preparation and Characterisation of Liposomes

Six liposome formulations were prepared in the following way: two formulations of unloaded liposomes (L1 and L2), two formulations of liposomes loaded with quercetin (L3 and L4) and two formulations of liposomes loaded with the PAH extract. The composition of the liposome formulations is presented below in Table 1.

Taking into account that the liposomes stability is limited due to the tendency of phospholipids to get oxidised, they were stored in well-sealed tubes at temperatures of 4–8 °C after preparation [50]. A liposome’s size affects its systemic distribution and clearance. The larger the liposome, the higher the risk of uptake and degradation by the reticuloendothelial system (RES). Based on the composition of phospholipids and the pH of the environment, liposomes can be negatively or positively charged or be neutral. The nature and density of the electrical charge on the surface of the liposomal membrane influences the stability and biodistribution kinetics, as well as the liposomal uptake by target cells [51].

Phosphatidylcholine-containing liposomes with a neutral charge have a lower chance of being recognised and destroyed by RES after administration. Phosphatidylserine-based liposomes are negatively charged and have a higher rate of endocytosis due to the recognition of the negatively charged surface by macrophage receptors. A small amount of lipids (cholesterol) was added to control the rigidity of the liposomal membrane and to reduce the instability caused by serum proteins binding to the membrane. Cholesterol is a common membrane constituent in biological systems, and it is involved in modulating the membrane fluidity, elasticity and permeability.

The size and zeta potential were measured using dynamic light scattering (DLS). The liposomes obtained with phosphatidylcholine and PAH extract (L5) were smaller than those obtained with phosphatidylcholine and quercetin (L3) but larger than those of bare liposomes (L1). The zeta potential is negative, and the lowest is quercetin liposomes (−33.16 mV), followed by liposomes with the PAH extract (−33.06 mV) and bare liposomes (−10.6 mV). Liposomes obtained with phosphatidylserine and PAH extract (L6) have the smallest dimensions, followed by liposomes with phosphatidylserine and quercetin (L4) but are larger than bare liposomes (L2). The zeta potential is negative, and the lowest are the bare liposomes (−29.8 mV), followed by those with the PAH extract (−26.63 mV) and the liposomes with quercetin (L4).

The liposome size can range from very small (25 nm) to large (2500 nm) vesicles. Furthermore, liposomes may have a two-layer membrane. The size of the vesicle is a parameter used to assess the distribution of liposomes, and the number of layers affects the amount of entrapped substance. The liposomes with sizes between 100 nm and 1000 nm are considered gigantic liposomes, according to the literature. Taking into account the results obtained and the demonstrations of other research on liposome technology, we can say that the liposomes obtained by the method of hydration of the lipid film applied in the preparation determined the obtaining of multilamellar, giant liposomes. Among the advantages of this type of liposome are an increased stability and ease of preparation [52]. As a result, several unilamellar vesicles formed on the inside of them, making a multilamellar structure of concentric phospholipid spheres separated by water layers. To make the following studies on liposomes more efficient, a separation of them was performed by filtration using a 450-nm microfilter, and a uniformity of size for them was obtained.

It was noted that the zeta potential was lower for phosphatidylcholine liposomes compared to those formed with phosphatidylserine, although the methods of obtaining them were identical. The distribution of the sizes of the obtained liposomes determined by DLS was shown in Figure 1.

The entrapment of quercetin in liposomes with phosphatidylcholine and cholesterol led to an increase in the liposome diameter. When phosphatidylcholine was used for the preparation, the diameter of the liposomes increased, being between 100 and 1000 nm with a percentage of approximately 80% for L3 and 77% for L5, compared to 78% for L1. The percentages of liposomes with a diameter of less than 100 nm also decreased from L1 (22%) to L3 (2%) and to L5 (1.5%). Even if, at L1, the diameter of the liposomes did not exceed 1000 nm, for the loaded liposomes, the diameter exceeded this value (L3—14% and L5—20%). In the preparation of liposomes with phosphatidylserine, the percentage of liposomes with a diameter of less than 100 nm increased from L2 (1%) to L4 (4%) and to L6 (25%). It can also be seen that the diameter of bare liposomes (L2) exceeded 1000 nm by 15%, this percentage decreasing to zero for loaded liposomes (L4 and L6), which means that the loading of liposomes with quercetin or PAH led to a decrease in their size. To make the following studies on liposomes more efficient, a separation of them was performed by filtration using a 450-nm microfilter and a uniformity of size for them was obtained. The particle size obtained was calculated from three sets of readings.

### 3.4. Liposome Morphology Evaluated by AFM

Liposome characterisation was performed by the AFM technique. Porous sphere morphology with micrometric dimensions is important in drug delivery systems. The main advantage of the AFM procedure is the possibility to operate with high resolution in the air or inside a fluid in real time and on a nanoscale [53].

The calculated values from the AFM images (average roughness (Sa), mean square root roughness (Sq), maximum peak height (Sp), maximum valley depth (Sv) and maximum peak-to valley height (Sy)) are shown in Table 3.

The results from Table 3 confirmed that samples L4 and L3 exhibited the highest roughness, followed by L5 and L6, while the lowest roughness value was observed in the cases of L2 and L1. The samples presented irregular agglomerated formations with the exception of the more regular-shaped in the case of samples L3 and L4. The round formations were more “compact” and closer to each other in sample L4. The base (L1 and L2) had a more irregular surface (Figure 2A,B) compared to the L5 and L6 samples that exhibited well-shaped surfaces indicated by the profile analysis (Figure 2E,F). The surface profile should not be confused with the overall roughness, which was observed by the valleys and heights of the sample, while the profile is an indicator of the surface appearance in a certain area. In our case, a low roughness indicated the absence of a compound on the surface that can or cannot create a layer. Together with higher roughness results, it can be concluded that L5 and L6 interacted with the introduced compounds. The same conclusion is viable for L3 and L4, which had the highest roughness and most visible agglomerations (Figure 2C,D).

The connection between the zeta potential and the roughness of the surface has been reported in several works [54,55,56]. It should be noted that the particle shape and morphology, together with pH, could also influence the zeta potential value [54,55]. In our study, the zeta potential has a negative value due to the accumulation of the charge carriers in certain areas as a result of the “backflow regions” that occurred as a consequence of different morphologies and, therefore, different asperities on the surface [55]. The area of the empty liposome formulas is smaller than that of the loaded ones, which means that both quercetin and the phytocomplex extracted from PAH were trapped. The morphological and structural characterisations of liposomes is important for their mode of action and gives an overview of their biological activity. Nanometrically modified surfaces promote adhesion by making possible the more natural bonding of bioactive nanoparticles to the cell membrane.

### 3.5. Liposome Entrapment Efficiency

Thanks to their bioactive substance content, liposomes guarantee the efficient transport to cells or tissues and the maintenance of a constant therapeutic concentration, reducing the risk of overdosing. Further investigations were needed in order to ascertain the utility of the formed liposomes. To this end, the degree of liposomal quercetin entrapment, as well as the total flavonoids from the PAH extract, were followed. The specimens showed good efficiency, as presented in Table 4. These results confirmed the importance of hydrating the lipid film through the method chosen by us.

We can specify the fact that the quercetin and PAH extract-loaded liposomes were stable after 60 days of storage at 4 °C, with minimal losses.

### 3.6. The Release of Flavonoids Entrapped in Liposomes Using a Franz Cell System

In this experiment conducted with the diffusion method, the PAH extract and quercetin entrapped in the two types of liposomes were released in a variable degree after 12 h. The results are shown in Figure 3. The quantity of PAH released was evaluated as a percent of the total released flavonoids expressed in QE. The total flavonoid quantity from the lipid formulations showed a slower release rate, as follows: L3 formulation—80.05 ± 3.45%, L4 formulation—82.25 ± 2.06%, L5 formulation—78.50 ± 3.16% and L6 formulation—77.20 ± 2.08%. The initial release rate was 70% in the first 6 h; after which, the concentration of the pharmacologically active ingredient was maintained. Moreover, it avoided uncontrolled substance release and allowed us to obtain a constant pharmacokinetic profile.

In order to investigate the mechanism of the in vitro flavonoid release, several kinetic models were analysed (zero order, first order, Higuchi and Hickson-Crowell). The models were compared using (R^2^) correlation. The data showed in vitro flavonoid release kinetics in concordance with the Higuchi model, which approximated the experimental measurements well. The resulting correlation quotient had a value of 0.9923 using KinetDS 3 rev 2010 software (version 3.0, Medyczna, Poland).

### 3.7. HUVECs Viability Assay

In order to establish the influence of free quercetin, the PAH extract and different formulations of liposomes with quercetin or the PAH extract on the HUVEC viability in the presence of Doxo, the MTS assay was performed. The cell viability was evaluated after exposure to different concentrations of quercetin and the L3 and L4 formulations (ranged between 0.001 μg GAE/mL and 100-µg GAE/mL) (Figure 4). For the PAH extract or L5 and L6 formulations, the concentrations used varied between 0.006 GAE/mL and 0.678 μg GAE/mL. Quercetin induced a slight viability decrease at doses above 0.1 µg/mL, while both formulations of liposomes with quercetin increased the cellular viability even at 100 µg/mL, especially the L3 formulation (Figure 4). The PAH extract led to significant changes in the cell viability above 0.33-µg GAE/mL compared to the control. The L5 formulation reduced the viability at doses exceeding 0.06 µg/mL (Figure 4), while L6 significantly decreased the cell viability, starting at 0.006 µg/mL (Figure 4). The decline of viability was dose-dependent in the presence of L5 and L6, while the L3 and L4 formulations increased the cellular viability of the HUVEC cells even at high doses. The different behaviours of the various formulations of liposomes depended on their composition and the compounds loaded. Starting from the viability test, the concentrations chosen for further experiments were 0.01-μg GAE/mL for all tested compounds.

### 3.8. Evaluation of Biological Activities on HUVECs Exposed to Doxo

In order to explore the mechanisms involved in the protection of quercetin, the PAH extract and different formulations of liposomes with quercetin and the PAH extract, an in vitro model of toxicity induced by Doxo administration on the HUVEC cell line was used. The toxicity of Doxo was assessed by malondialdehyde in cell lysates, cytokine IL-6 and caspases levels in the cellular medium and, also, by the quantification of DNA lesions and transcription factors in endothelial cells.

MDA increased in cell lysates after Doxo exposure (*p* < 0.001) and diminished significantly both after PAH extract and quercetin administration (*p* < 0.001) compared to the control cells (Figure 5A). The two formulations of liposomes with quercetin (L3 and L4) reduced the MDA levels, the best results being obtained after the L3 treatment (*p* < 0.001) if the results were compared to those obtained after the L4, L5 and L6 administrations. The liposomes loaded with the PAH extract decreased the lipid peroxidation (*p* < 0.001) without a significant difference between PAH and two forms of liposomes with the PAH extract (L5 and L6) (Figure 5A).

The inflammation induced by Doxo was assessed by IL-6 secretion in the supernates of cells exposed to quercetin, PAH extract and different liposomes formulations. The IL-6 levels increased significantly in the supernates of cells treated with Doxo and diminished after the PAH extract (*p* < 0.001), L3 (*p* < 0.001), L4 (*p* < 0.001), L5 (*p* < 0.001) and L6 (*p* < 0.05) administrations. The anti-inflammatory effect was comparable between the PAH extract and L3 formulation (Figure 5B).

To evaluate the effect of quercetin, the PAH extract and various formulations of liposomes on apoptosis, caspases-3, -8 and -9 were evaluated in the supernates of cells exposed to Doxo (Figure 6). Thus, Doxo administration increased the apoptotic index (*p* < 0.001), while the PAH extract diminished the caspase-9 (*p* < 0.001) and -8 (*p* < 0.01) levels. The liposomes loaded with PAH extract reduced caspases-3 and -9, especially the L6 formulation (*p* < 0.001). Quercetin administered in cells incubated with Doxo significantly decreased all caspases in the supernates (*p* < 0.001). The same effect was obtained after the L3 liposomes prepared with quercetin (*p* < 0.001).

In our experimental design, Doxo exposure significantly induced Nrf2 (*p* < 0.05) and NF-kB activation (*p* < 0.01) without affecting the DNA in the administered dose (Figure 7A–C). The expression of constitutive NF-kB and the phosphorylated form pNF-kB quantified by Western blotting showed NF-kB activation after the L3 treatment (*p* < 0.01), while the L4 and L5 formulations downregulated the pNF-kB/NF-kB ratio (*p* < 0.05). The DNA lesions quantified by γH2AX expression in cell lysates were reduced after quercetin (*p* < 0.001) and the liposomes loaded with quercetin, especially L4 (*p* < 0.001), and, also, after PAH and the liposomes loaded with PAH (L5: *p* < 0.001) (Figure 7D). The best protection was found in the cells treated with the L4 formulation.

In the last years, there have been numerous studies on Doxo cytotoxicity, particularly on cardiomyocytes, but the mechanisms responsible are not completely elucidated. The cytotoxic effects were related especially to easy diffusion in the intracellular environment, the intercalation in DNA and the inhibition of topoisomerase II with the reduction of cell division and growth [57,58]. Moreover, Doxo induced mitochondrial dysfunction due to the accumulation of drugs inside the mitochondria and generated considerable quantities of reactive oxygen species (ROS) [59,60]. ROS production is responsible for macromolecules oxidation, including myofibrillar proteins and the induction of muscular dysfunction [18,61].

Doxo is converted to a radical by various mechanisms, particularly in the presence of iron released from aconitase or ferritin [62,63]. When antioxidant enzymes are exceeded, the superoxide anion forms peroxynitrite, cytotoxic compounds for lipids, proteins and nucleic acids [64]. The mitochondrial damage triggers the activation of caspases and downregulates the B-cell lymphoma (Bcl)-2 protein expression [65], key factors involved in apoptosis. Additionally, Doxo can directly damage the DNA by the inhibition of topoisomerase II, an enzyme implicated in DNA repair, and subsequently, DNA aggregation and breakage occur, especially after high doses.

Moreover, in the systemic administration of Doxo, the first contact is with the vascular endothelium, where the drug can generate endothelial dysfunction, hyperpermeability due to the damage of the tight junction between endothelial cells, haemorrhage, atherosclerosis and restenosis [19,26]. Therefore, the strategy that provides the protection of the vascular endothelium can improve the cardio protection and reduce the risk of developing cardiomyopathy after Doxo administration.

The antioxidant effect of the tested compounds, especially of liposomes loaded with free quercetin and the PAH extract, can afford a good protection against oxidative stress-induced toxicity and, accordingly, against the activation of cellular pathways involved in inflammation and cell death. In our study, both free quercetin and liposomes loaded with quercetin inhibited oxidative stress and nonspecific inflammation and reduced DNA lesions and apoptosis. In parallel, these compounds diminished Nrf2 and NF-kB expressions, particularly the L4 formulation. The PAH extract and liposomes loaded with the PAH extract exerted an antioxidant activity, diminished the secretion of proinflammatory cytokine and γH2AX formation and reduced extrinsic apoptosis. L5 and L4 inhibited Nrf2 and NF-kB activation, while L3 increased the pNF-kB expression. These effects were correlated with the high content of flavonoids or phenolic acids in the extract and a good antioxidant capacity evaluated by DPPH, ABTS, FRAP and CUPRAC.

Our results on the antioxidant activities of the PAH extract were in agreement with other findings found in the in vitro and in vivo experiments. Thus, Wu et al. [66] found that *Polygonum orientale* flower extract had a protective effect on H₂O₂-induced HUVEC cell injury by the inhibition of MDA formation and downregulation of Bcl-2 expression, while *Polygonum multiflorum* increased the activities of superoxide dismutase and glutathione peroxidase [67]. Doxo reduced the activities of catalase and superoxide dismutase 1 and 2, while the *Polygonum maritimum* extract improved the activity of these enzymes in cancer cells [68]. The anti-inflammatory effect of other species of Polygonum was demonstrated by Bralley et al. [69] on a mouse ear inflammation model induced by the topical application of 12-O-tetradecanoylphorbol-13-acetate (TPA). The Polygonum extract reduced the oedema and infiltration with neutrophils in the treated ear, the effect comparable with Indomethacin but without side effects. Therefore, the entrapment of the PAH extract in liposomes as carriers in order to increase the cell penetration and protection efficacy is a good strategy for the delivery of bioactive compounds to target cells.

It is known that chemotherapy with Doxo can induce dose-dependent endothelial dysfunction due to the reduction of vasodilators bioavailability—in particular, nitric oxide—in parallel with increasing the vasoconstrictor agents [70]. In this process, NF-kB plays an important role as the transcription factor involved in cell proliferation, inflammation and differentiation as a response to oxidative stress [71]. Generally, free radicals generated by Doxo administration determine the NF-kB translocation into the nucleus and upregulation of different pathways [72]. El-Agamy et al. [73] demonstrated that Nrf2 and NF-kB inhibition by Pristimerin reduced the cardiotoxicity triggered by Doxo, while Xu et al. [71] found high expressions of p53 and phosphorylated p53 in Doxo heart rat vessels compared to untreated vessels. These findings confirmed the translocation of p53 into the nucleus after a Doxo treatment and NF-kB pathway activation and, also, its involvement in cardiac fibrosis [74]. It seems that the relationship between the activation of NF-kB and apoptosis is a complex, dual one, in which NF-kB can promote or inhibit apoptosis, depending on the type of cells or the nature of the stimuli [75]. Thus, in various cancer cells, Doxo-induced apoptosis is associated with the inhibition of NF-kB activation, suggesting the antiapoptotic role of NF-kB [76]. In our study, liposomes loaded with quercetin (L3) activated NF-kB, while oxidative stress and inflammation were inhibited. Moreover, L3 diminished the activity of the caspases, suggesting its antiapoptotic, antioxidant and anti-inflammatory properties. Probably, NF-kB activation induced by different stimuli upregulated the expression of genes involved in the resistance to cell death [77], including antiapoptotic proteins [78] and antioxidant enzymes. This mechanism was demonstrated both in different normal cells, such as primary rat and human fibroblasts, Jurkat T cells and in tumour cells of bladder cancer or breast carcinoma [79,80]. These data attested the dual role of NF-kB in promoting cell survival depending on the context, cells and experimental conditions [81].

It was demonstrated that a redox imbalance was associated with the nuclear translocation of Nrf2 and targeting genes of the antioxidant response element (ARE) region, such as heme oxygenase-1 (HO-1), NADPH dehydrogenase quinone 1 (NQO1) and g-glutamyl cysteine ligase [82]. Nrf2 activation reduced the inflammatory response and the subsequent inhibition of NF-kB signalling, while its inhibition increased the NF-kB activation and production of proinflammatory cytokines [83]. In our experiment, both quercetin and liposomes with quercetin, as well as the PAH extract and liposomes with the PAH extract, significantly diminished the expression of Nrf2 (*p* < 0.001) in cells exposed to Doxo in parallel with the inhibition of oxidative stress and inflammation.

The Nrf 2 activation in cancer cells reduced the chemotherapeutic toxicity by improving the antioxidant enzyme system and dropping the apoptosis but, at the same time, mediated the chemoresistance and allowed tumour cells to survive [84]. Unlike the dual effect of Nrf2 in tumour cells, in normal ones, the downregulation of Nrf2 is protective and ameliorates the Doxo-mediated cardiac and vascular toxicity triggered by oxidative stress. Doxo enhanced the Nrf2 expression and upregulated the superoxide dismutase, catalase and glutathione peroxidase activities. In our study, all the tested liposomes reduced the Nrf2 expression induced by Doxo and maintained low levels of ROS, suggesting their beneficial effect in vascular protection.

Additionally, oxidative stress is associated with the loss of mitochondrial membrane potential, cytochrome C release and apoptosis induction. Therefore, the agents that reduce oxidative, inflammation and apoptosis can be beneficial in cardio and vascular protection. Both quercetin and liposomes formulation with quercetin, particularly L3, diminished the caspases, while only liposomes loaded with the PAH extract had protective properties against apoptosis triggered by Doxo, especially extrinsic apoptosis. These results were consistent with the other literature data. Accordingly, the cardioprotective effect of quercetin was demonstrated in an in vitro experimental model on H9c2 cell lines subjected to hypoxia/reoxygenation. Thus, a pretreatment with 10–16 µM of quercetin resulted in a beneficial effect on cardiomyocytes by c-Jun N-terminal kinase and p38 mitogen-activated protein kinase inhibition and the modulation of proapoptotic protein Bcl-2 and Bax expressions [85]. Quercetin encapsulated in poly (lactic-co-glycolic acid) nanoparticles exerted a better cardioprotective action compared to free quercetin on the same experimental model due to decrease in the ROS production and by the improvement of the mitochondrial function and ATP synthesis [86].

Co-delivered quercetin and Doxo by using a nanocarrier composed of methoxy poly(ethylene glycol) and poly(d, l-lactide-co-glycolide) (mPEG–PLGA NPs) exerted good protection on normal vascular endothelial cells and enhanced the antitumour effect of Doxo [87]. Quercetin increased the Nrf2 mARN expression and maintained the integrity of the cell membrane and, also, the histological abnormalities induced by Doxo exposure in rats [88]. These results confirmed the superior protective properties after the inclusion of quercetin in nanocarriers. It is known that quercetin is an effective antioxidant with a protective effect on ROS-induced damage [89], but due to a poor aqueous solubility and poor dissolution in gastrointestinal fluids, the intestinal absorption is reduced and its effect much diminished [90]. To improve the solubility, quercetin was added to liposomes that were able to carry and transport it due to the lipid bilayer and aqueous core [91]. In our study, the incorporation of quercetin into liposomes increased their antioxidant and anti-inflammatory efficacy, especially those formulated with phosphatidylcholine (L3), and inhibited apoptosis. In addition, quercetin, the PAH extract and liposomes loaded with two compounds inhibited the γH2AX foci formation enhanced by Doxo, suggesting its protective activity on DNA lesions. ɣH2AX is a marker that reflects the persistence of irreparable DNA double-stranded brakes after cell exposure to different genotoxic agents [90] and is frequently used to quantify DNA damage. In our experiment; irreversible DNA damage was reversed efficiently by all the tested compounds, and the best protection was exerted by the L4 liposomes.

## 4. Conclusions

In conclusion, this study provided evidence of the mechanisms involved in the endothelial toxicity of Doxo and brought up arguments for the endothelial protection of liposomes loaded with quercetin and the PAH extract. Doxo administration decreased the cell viability and induced oxidative stress, inflammation, DNA lesions and apoptosis in parallel with the activation of Nrf2 and NF-kB. Endothelial protection by the tested compounds was different depending on the liposomes formulation and the active substance used for loading. Thus, the best protection was exerted by liposomes prepared with phosphatidylcholine, especially those entrapped with quercetin. Both formulations of liposomes loaded with quercetin inhibited oxidative stress and nonspecific inflammation and reduced the DNA lesions and apoptosis. Additionally, these compounds diminished the Nrf2 and NF-kB expressions—principally, the L4 formulation—while L3 increased the NF-kB activation as a mechanism of cell survival.

The PAH extract and liposomes loaded with the PAH extract exerted antioxidant and anti-inflammatory activities, diminished the DNA lesions and inhibited extrinsic apoptosis and the activation of the transcription factors but to a lesser extent than quercetin and the liposomes with quercetin. These different results can be explained by different compositions and different behaviours of the liposomes in vitro. Certainly, the factors involved are varied and include how liposomes reach and act in a living cell, as well as the mechanisms by which they interfere. Therefore, further investigations are necessary for the evaluation of the biological effects of the extract and liposomes with the extract on endothelial cells and cardiomyocytes to maximise their benefits in cardio protection and vascular protection.

## Figures and Tables

**Figure 1 pharmaceutics-13-01418-f001:**
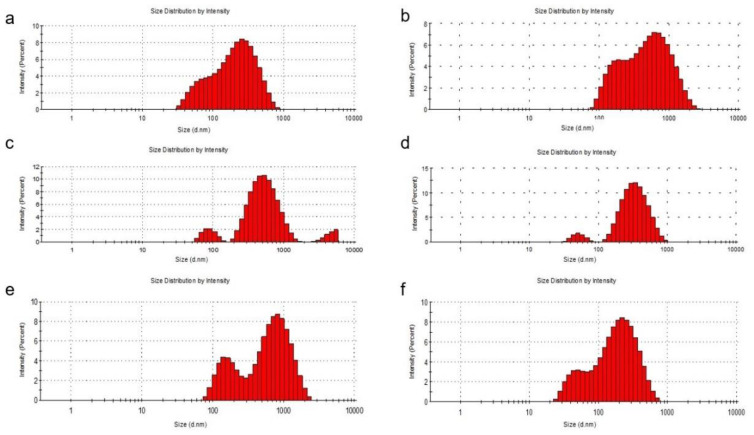
DLS analysis of the liposomal systems. Particle size distribution of the L1 (**a**), L2 (**b**), L3 (**c**), L4 (**d**), L5 (**e**) and L6 (**f**) formulations.

**Figure 2 pharmaceutics-13-01418-f002:**
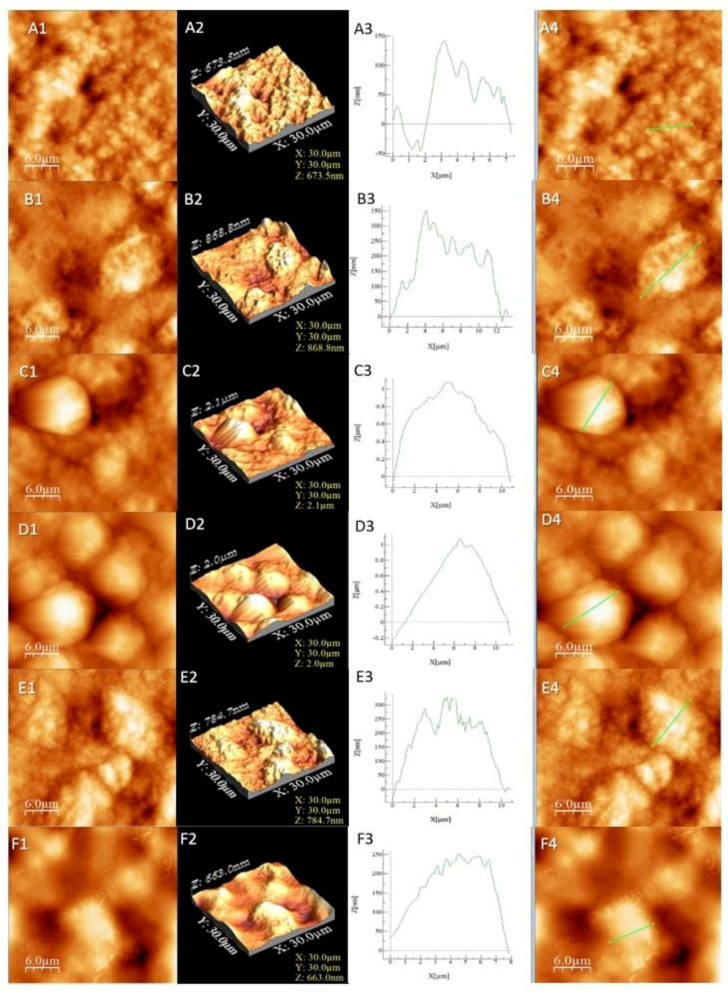
AFM images for the studied liposomes. The letters correspond to (**A**) L1, (**B**) L2, (**C**) L3, (**D**) L4, (**E**) L5 and (**F**) L6. The numbers correspond to (**1**) AFM image, (**2**) 3D image, (**3**) profile of the sample and (**4**) area selected for profiling.

**Figure 3 pharmaceutics-13-01418-f003:**
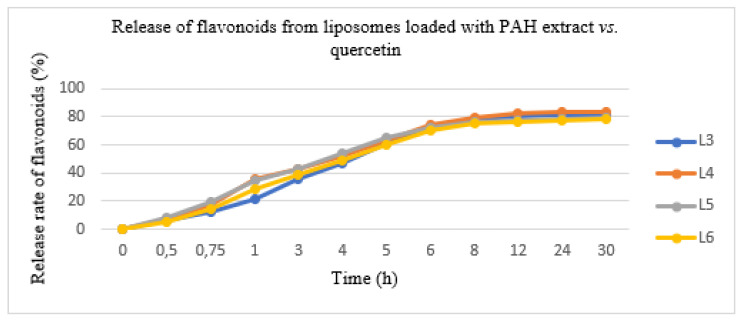
The release of flavonoids from liposomes loaded with the PAH extract (L5 and L6) vs. liposomes loaded with quercetin (L3 and L4).

**Figure 4 pharmaceutics-13-01418-f004:**
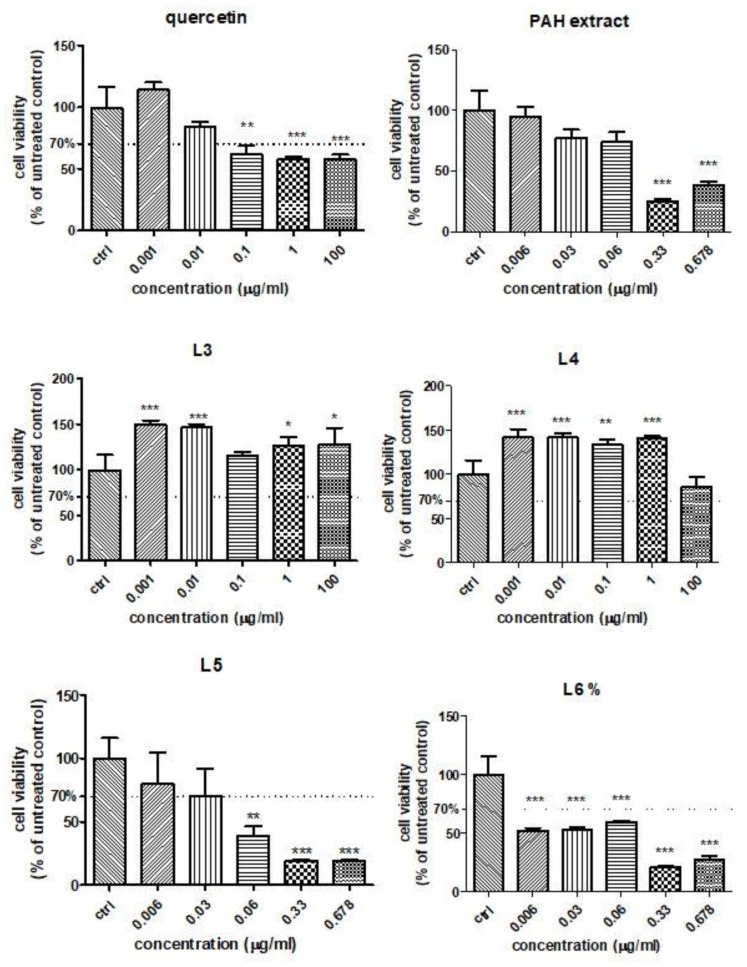
Cell viability of HUVECs treated with quercetin, PAH extract and different liposome formulations with quercetin and the PAH extract. HUVECs were exposed to quercetin and L3 and L4 formulations in concentrations ranging between 0.001-µg GAE/mL and 100-µg GAE/mL and to the PAH extract and L5 and L6 formulations in concentrations between 0.006-µg GAE/mL and 0.678-µg GAE/mL (data are presented as the mean of OD540 ± SD; *n* = 3 for each sample). * *p* < 0.05, ** *p* < 0.01 and *** *p* < 0.001 vs. control, untreated cells.

**Figure 5 pharmaceutics-13-01418-f005:**
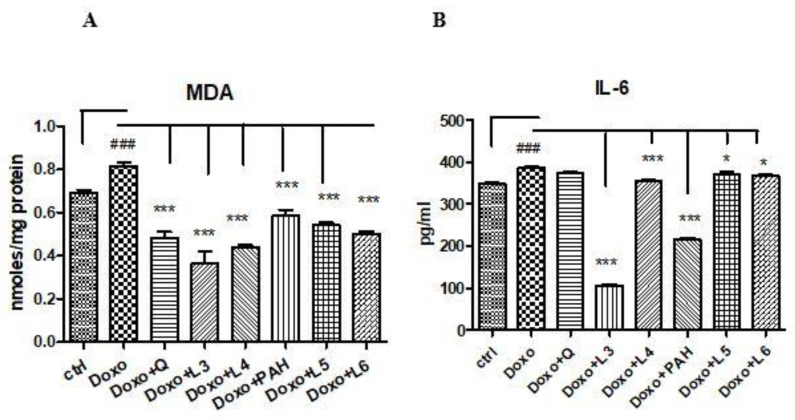
Malondialdehyde levels in the cell lysates, and IL-6 secretion in the supernates of the HUVECs exposed to Doxo and quercetin, the PAH extract and different formulations of liposomes. (**A**) MDA increased significantly after Doxo administration and diminished in the cells treated with quercetin (*p* < 0.001), the PAH extract (*p* < 0.001), L3 and L4 with quercetin (*p* < 0.001) and L5 and L6 with the PAH extract (*p* < 0.001). (**B**) IL-6 secretion increased after Doxo exposure (*p* < 0.001) and diminished in the supernates of the cells treated with L3, L4 (*p* < 0.001), L5 and L6 (*p* < 0.05). The statistical significance of the differences between the treated and control groups was evaluated with one-way ANOVA, followed by Dunnett’s multiple range test. Data are presented as the mean and SD of triplicate samples. ^###^ vs. control and * *p* < 0.05; *** *p* < 0.001 vs. Doxo-treated cells.

**Figure 6 pharmaceutics-13-01418-f006:**
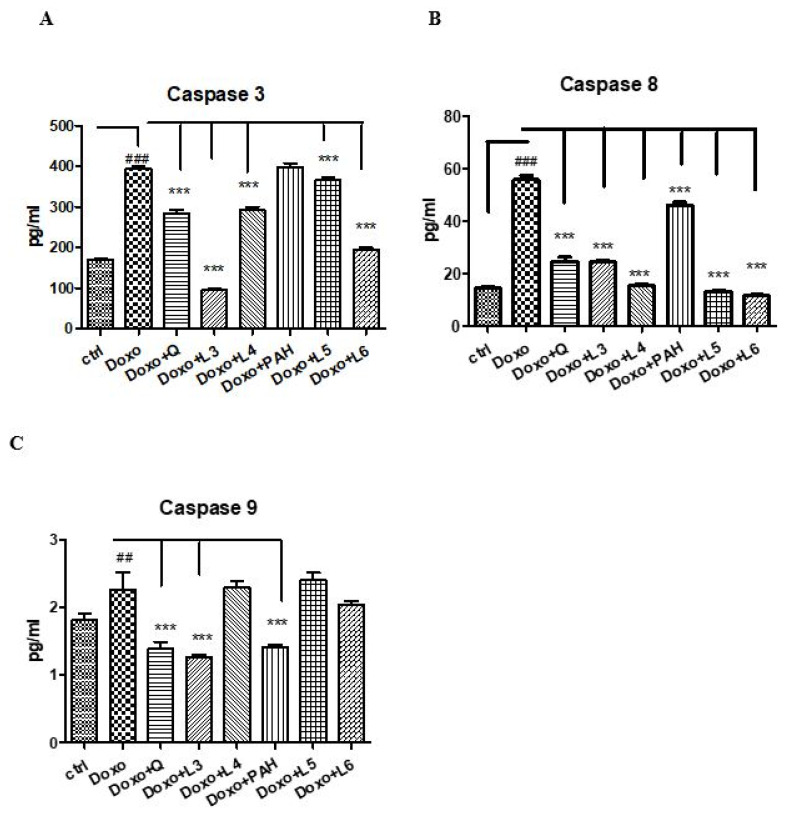
Caspases-3, -8 and -9 in the supernates of HUVECs exposed to Doxo and quercetin, the PAH extract and different liposomes formulations. Caspases-3 (**A**), -8 (**B**) and -9 (**C**) increased significantly after Doxo exposure and diminished both in cells treated with quercetin (*p* < 0.001) and L3 (*p* < 0.001). The PAH extract significantly reduced caspase-8 and caspase-9 (*p* < 0.001), while L5 and L6 decreased the caspase-3 and caspase-8 levels (*p* < 0.001). The statistical significance of the difference between the treated and control groups was evaluated with one-way ANOVA, followed by Dunnett’s multiple range test. Data are presented as the mean and SD of triplicate samples. ^##^
*p* < 0.01; ^###^
*p* < 0.001 vs. control and *** *p* < 0.001 vs. Doxo-treated cells.

**Figure 7 pharmaceutics-13-01418-f007:**
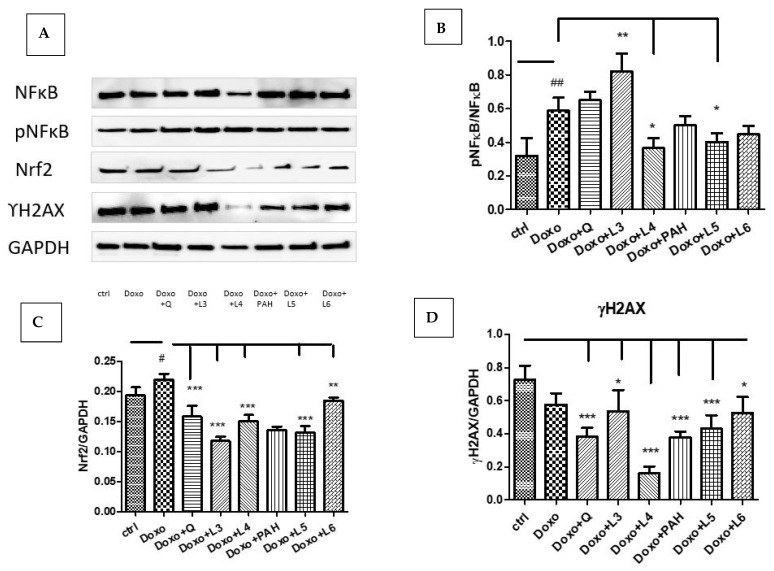
NF-kB, pNF-kB, Nrf2 and γH2AX expressions in HUVEC lysates after exposure to Doxo and the treatments with quercetin; the PAH extract and the L3, L4, L5 and L6 formulations. Representative images of immunoblotting for NF-kB, pNF-kB, Nrf2, γH2AX and GAPDH in the HUVECs treated with quercetin, the PAH extract and liposomes loaded with quercetin and the PAH extract are shown in the upper panel (**A**), and the results of the statistical analysis for the ratio of Nrf2, γH2AX and GAPDH expressions are in the lower panels (**C**,**D**). For NF-kB and pNF-kB, the results were expressed as pNF-kB divided by the total NF-kB (**B**). The statistical significance of the differences between treated and control groups was evaluated with one-way ANOVA, followed by Dunnett’s multiple range test; ^#^
*p* < 0.05; ^##^
*p* < 0.01 vs. control cells and * *p* < 0.05; ** *p* < 0.01; *** *p* < 0.001 vs. Doxo-treated cells.

**Table 1 pharmaceutics-13-01418-t001:** The composition of the liposome formulations.

Liposome Composition (mg)	L1	L2	L3	L4	L5	L6
Phosphatidylcholine (PC) (mg)	80	-	80	-	80	-
Phosphatidylserine (PS) (mg)	-	80	-	80	-	80
Sodium cholate (CoN) (mg)	20	20	20	20	20	20
Cholesterolum (CHO) (mg)	2.5	2.5	2.5	2.5	2.5	2.5
Quercetin (QE) (mg)	-	-	10	10	-	-
*Polygonum aviculare herba* extract (PAH) (mg)	-	-	-	-	100	100

**Table 2 pharmaceutics-13-01418-t002:** The contents in the phenolic compounds measured in the PAH extract.

Bioactive Compounds	λ_max_ (nm)	R_T_ (min)	PAH Extract mg/100 mg DW
Coumarins			
7-methoxycoumarin	337	14.223	0.425 ± 13.782
Bergapten	270	27.958	N.D.
Isopimpinelline	270	18.706	N.D.
Flavonoids			
Delphinidin 3-rutinoside chloride	275	6.156	1.034 ± 6.326
Diosmin	337	17.637	N.D.
Hyperoside	360	15.333	N.D.
Luteoline 7–glucoside	360	14.442	N.D.
Rutin	360	15.780	0.094 ± 15.766
Quercetin	360	19.543	0.603 ± 19.543
Phenolic acids			
Caffeic acid	326	7.445	0.092 ± 7.433
Chlorogenic acid	326	6.705	0.076 ± 6.528
Ellagic acid	254	16.862	0.393 ± 16.630
Ferulic acid	326	10.799	N.D.
Gallic acid	254	2.976	0.026 ± 3.075
Rosmarinic acid	326	16.050	2.65 ± 16.089
Trans p-coumaric acid	326	10.198	0.056 ± 10.108
Stilbenes			
Trans resveratrol	337	15.768	N.D.

**Table 3 pharmaceutics-13-01418-t003:** Roughness results from the AFM images.

Sample Name	Area (Ironed Area) (µm^2^)	Sa (µm)	Sq (µm)	Sp (µm)	Sv (µm)	Sy (µm)	Roughness
L1	910	0.070	0.090	0.338	−0.334	0.673	+
L2	912	0.099	0.126	0.473	−0.395	0.868	+
L3	929	0.234	0.312	1.193	−0.902	2.095	+++
L4	931	0.301	0.404	1.320	−1.034	2.354	+++
L5	917	0.101	0.126	0.394	−0.389	0.784	++
L6	905	0.087	0.108	0.368	−0.294	0.663	++

Sa—average roughness, Sq—mean square root roughness, Sp—maximum peak height, Sv—maximum valley depth and Sy—maximum peak-to-valley height.

**Table 4 pharmaceutics-13-01418-t004:** The liposome entrapment efficiency at three different storage intervals.

Liposome Specimens	*EE*% at the Time of Preparation	*EE*% at 10 Days	*EE*% at 30 Days	*EE*% at 60 Days
L3	85 ± 2.01	84.6 ± 2.14	83.8 ± 0.97	83 ± 1.91
L4	88.4 ± 1.91	88.2 ± 2.03	88 ± 0.16	87.8 ± 0.75
L5	79 ± 1.07	78.8 ± 0.27	78.7 ± 0.64	78.5 ± 1.25
L6	81 ± 2.12	80.8 ± 1.24	80.6 ± 1.18	80.5 ± 1.54

*EE*%—entrapment efficiency.

## Data Availability

No applicable.

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
