# Peer review of "Comparative Study of the Pharmacological Properties and Biological Effects of Polygonum aviculare L. herba Extract-Entrapped Liposomes versus Quercetin-Entrapped Liposomes on Doxorubicin-Induced Toxicity on HUVECs"

_pharmaceutics, 2021, doi:10.3390/pharmaceutics13091418_

Round 1
Reviewer 1 Report
The work described in the present manuscript is consistent with the scope of the journal. The author's research work “Pharmacological properties and biological effects of Polygonum 2 aviculare L. herba extract-entrapped liposomes on Doxorubicin- 3 induced toxicity on HUVECs” has discussed in detail the development and detailed in vitro characterization and evaluation of the liposomes. This work is methodically carried out and scientifically correct. There are major issues that the authors can address to improve their manuscript before acceptance for publication.
What is the difference between L3 and L4 in the abstract? Authors need to clarify at first instance then use short form or abbreviated form.
What was the purpose of making liposomal formulation of quercetin? Is this the first time reported?
There is no literature present on quercetin and its formulation aspects in the introduction.
Why authors have added 30% ethanol in vitro release study
Please include the IC50 for formulations.
Authors should modify figure 4 including % cell viability/cytotoxicity in y-axis.
The authors did not show any experiments in which they used blank formulation. liposomes themselves could induce an unwanted cytotoxic effect on cells. MTT assay should be performed also with the blank formulation.
Authors should modify the title of the paper as quercetin is showing better results but not mentioned in the title of the paper.
Discussion of the results is laconic; the author needs to improve the discussion compared to previous literature.
Authors can include studies on apoptotic activities and cell cycle analysis to further prove the role of present formulation in anticancer activity if possible.
Author Response
The work described in the present manuscript is consistent with the scope of the journal. The author's research work “Pharmacological properties and biological effects of Polygonum 2 aviculare L. herba extract-entrapped liposomes on Doxorubicin-induced toxicity on HUVECs” has discussed in detail the development and detailed in vitro characterization and evaluation of the liposomes. This work is methodically carried out and scientifically correct. There are major issues that the authors can address to improve their manuscript before acceptance for publication.
- What is the difference between L3 and L4 in the abstract? Authors need to clarify at first instance then use short form or abbreviated form.
Response: The abbreviations L3, L4, L5, L6 represent the names of the liposomal formulations studied. The difference between L3 and L4 was due to the lipide type used for preparation. L3 was prepared using phosphatidylcholine and L4 phosphatidylserine (see in Abstract line 28 and 29)
- What was the purpose of making liposomal formulation of quercetin? Is this the first time reported?
Response: Quercetin containing liposomes have been studied. We’ve selected them as a reference in this study. This comparison was chosen due to quercetin-rich content of Polygonum herba extract and because quercetin has shown protective endothelial properties in Doxo exposure.
“Quercetin was chosen for comparison because it is an important component of the tested extract and because it has demonstrated endothelial-protective properties in doxorubicin-induced toxicity [(Hanan A. Henidi, Fahad A. Al-Abassi, Mohamed A. El-Mosselhi, Hany M. El-Bassossy, Ahmed M. Al-Abd. Despite blocking Doxorubicin-induced vascular damage, Quercetin ameliorated antibreast cancer activity. Oxid Med Cell Longevity ID 8157640, 2020)]. On isolated aortic rings incubated with Doxo quercetin restored the normal vascular contraction and relaxation impaired by Dox exposure and diminished the ROS generation. In addition, quercetin inhibited the ERK/MAP-kinase activation in ROS-induced cardiomyopathy [M. Kyaw, M. Yoshizumi, K. Tsuchiya, K. Kirima, and T. Tamaki, “Antioxidants inhibit JNK and p38 MAPK activation but not ERK 1/2 activation by angiotensin II in rat aortic smooth muscle cells,” Hypertension Research, vol. 24, no. 3, pp. 251–261, 2001].”
- There is no literature present on quercetin and its formulation aspects in the introduction.
Response: We have included aspects on the protective effect of quercetin on endothelium exposed to Doxo (see Introduction page 3, lines 111-114)
- Why authors have added 30% ethanol in vitro release study.
Response: Due to the fact that the in vitro release study was developed through a synthetic membrane, and the water solubility of active compounds is reduced, in order to maintain them in the receptor solution, we’ve added 30% ethanol, according to reference no. 39.
- Please include the IC50 for formulations.
Response: The IC50 value is the concentration required to inhibit radical formulation by 50%. The IC50 of Polygonum aviculare L. extract are 51.18 µg/ml.
- Authors should modify figure 4 including % cell viability/cytotoxicity in y-axis.
Response: The figure 4 was changed and the cell viability on y-axis was expressed as % of untreated control.
- The authors did not show any experiments in which they used blank formulation. liposomes themselves could induce an unwanted cytotoxic effect on cells. MTT assay should be performed also with the blank formulation.
Response: The experiments with blank formulation of liposomes (data not shown) displayed that both of them are toxic only at high concentrations (at values that exceed 0.11 mg/ml lipids respectively 100 µg/ml). For our experiments concentrations below 100 µg/ml were used as following.
- Authors should modify the title of the paper as quercetin is showing better results but not mentioned in the title of the paper.
Response: The title of the manuscript was changed as following: Comparative study of the pharmacological properties and biological effects of Polygonum aviculare L. herba extract-entrapped liposomes versus quercetin-entrapped liposomes on Doxorubicin-induced toxicity on HUVECs
- Discussion of the results is laconic; the author needs to improve the discussion compared to previous literature.
Response: Discussion section was reorganized and improved with some literature data regarding the effect of quercetin in endothelium protection and cardio-protection as following:”
“Co-delivered quercetin and Doxo by using a nanocarrier composed of methoxy poly(ethylene glycol) and poly(D, L-lactide-co-glycolide) (mPEG–PLGA NPs) exerted a good protection on normal vascular endothelial cells and enhanced the antitumor effect of Doxo [87]. Quercetin increased Nrf2 mARN expression and maintained the integrity of cell membrane and also the histological abnormalities induced by Doxo exposure in rats [88]”.
- Authors can include studies on apoptotic activities and cell cycle analysis to further prove the role of present formulation in anticancer activity if possible.
Response: The aimed of our study was to evaluate the protective effects of Polygonum aviculare L. herba (PAH)-entrapped liposomes and quercetin-entrapped liposomes on Doxorubicin (Doxo)-induced toxicity in HUVECs. The anticancer effect of tested compounds may be the subject of another study. This it is a very good idea and worth following in a future study.

Reviewer 2 Report
This manuscript submitted by Mariana Mureşan et al. (pharmaceutics-1346739) was described, some biological effects of Polygonum aviculare L. herba extract-entrapped liposomes and the improvement possibility of them on the cytotoxicity induced by Doxo.
I thought that this manuscript was a value in clue for new chemotherapy, however there were too long general informations in the discussion section (L639-671, L699-714, L724-728, L733-742). These explanations should be shorted, and should be moved into introduction (or be omitted).
The expression level of gammaH2AX was quantified by western blot, but I could be not able to think same result in 7A and 7D. The most decrease in photograph (Figure 7A) was lane Doxo+L4. But in figure 7D, the one was lane Doxo+L3, although internal control level (GAPDH) was similar. Furthermore, please add legend in figure 7A photograph in order to distinguish each lane.
The most amount contents in Polygonum aviculare L. herba extract is rosmarinic acid. In this paper, quercetin was used as positive control. Did the effect of rosmarinic acid examine (consider)?
In HUVECs viability assay, the composition and compound loaded was explained as a reason for different behavior. However, your results of L5 and L6 liposome such as EEV and releasing rate, was very similar. So, other reasons should be speculated.
The particle size was showed in Figure 1. Why did all liposome exist in two peaks? In L456, there was the description of filtration. If you used prepared liposome after filtration, DLS analysis should be re-performed using filtered liposomes.
In Figure 3, since all graphs were used same symbol and same line shape, each graph was not able to distinguish. Please re-draw with different symbols.
In Figure 2, cell viability was showed, but it was generally represented as percentage vs control. At least, scale (OD in y-axis) should be unified (Same scale 0.0-0.6).
Physical characters of liposome would be like to list in new table or to insert into Table 3.
In flavonoid releasing experiment, several kinetic models have been analyzed. Please show detailed results or omit these descriptions.
Following are minor points:
The company name “Schimadzu” in L146 and L343 is a mistake of “Shimadzu”.
The section number “3.2” in L232 is a mistake of “2.3”.
What is means “CDR(%)”? There was no abbreviation.
The cell density “104/cm2” is a mistake of “104/cm2”.
In L304, two “microg” should be replace to “µg”.
The antioxidant capacity of DPPH was explained to % value. How was the value calculated? And, what concentration was used of PAH in this assay?
“IC50” in L383 is a mistake of “IC50”.
Lipid name “PC” in L445 should be replaced into “phosphatidylcholine”.
The section title ” 3.6. The release of flavonoids entrapped in liposomes using a Franz cell system” makes it bold font.
The releasing rate was showed in text (L527-529). What time were these values exhibited?
“in vitro” descriptions in L535, L537, L750, and L573 should be italic font.
The word “polyphenols/ml” in L553 should be replaced into “GAE/ml”.
I seem to become more excellent paper, if the authors mention the above points.
Therefore, I regard that this manuscript is needed to revise for publishing in this journal.
Author Response
Reviewer R2
- I thought that this manuscript was a value in clue for new chemotherapy, however there were too long general information in the discussion section (L639-671, L699-714, L724-728, L733-742). These explanations should be shorted, and should be moved into introduction (or be omitted).
Response: The manuscript was shorted as you recommended. Thanks for your suggestion.
- The expression level of gammaH2AX was quantified by western blot, but I could be not able to think same result in 7A and 7D. The most decrease in photograph (Figure 7A) was lane Doxo+L4. But in figure 7D, the one was lane Doxo+L3, although internal control level (GAPDH) was similar. Furthermore, please add legend in figure 7A photograph in order to distinguish each lane.
Response: Thank you for your observation. The quantification of the gammaH2AX expression was corrected. The lowest level of gammaH2AX was obtained for L4. In addition, in order to distinguish each lane, a figure legend for 7A was added.
- The most amount contents in Polygonum aviculare L. herba extract is rosmarinic acid. In this paper, quercetin was used as positive control. Did the effect of rosmarinic acid examine (consider)?
Response: We’ve also observed this issue. This is a good idea for another study. We have chosen quercetin due to its demonstrated role in cardio-protection and endothelium protection.
- In HUVECs viability assay, the composition and compound loaded was explained as a reason for different behavior. However, your results of L5 and L6 liposome such as EEV and releasing rate, was very similar. So, other reasons should be speculated.
Response: The behavior of liposomes in vitro in terms of the release of the active substance were similar but their effects on HUVECs cells was different. Certainly, the factors involved are varied and include how liposomes reach and act in the living cell as well as the mechanisms by which they interfere. Therefore, further investigations are necessary for the evaluation of the biological effects of the extract and liposomes with the extract on endothelial cells and cardiomyocytes to maximize their benefit in cardioprotection and vascular protection.
- The particle size was showed in Figure 1. Why did all liposome exist in two peaks? In L456, there was the description of filtration. If you used prepared liposome after filtration, DLS analysis should be re-performed using filtered liposomes.
Response: Following the DLS analysis, we’ve noticed the variation in size of the liposomes. Filtration was carried out after the DLS analysis, because we wanted same sized liposomes for the assays that followed.
- In Figure 3, since all graphs were used same symbol and same line shape, each graph was not able to distinguish. Please re-draw with different symbols.
Response: The Figure 3 was re-draw with different symbols in order to clearer.
Figure 3 – The release of flavonoids from liposomes loaded with PAH extract (L5 and L6) vs. liposomes loaded with quercetin (L3 and L4)
- In Figure 2, cell viability was showed, but it was generally represented as percentage vs control. At least, scale (OD in y-axis) should be unified (Same scale 0.0-0.6).
Response: The figure 4 was represented as % of untreated control (y-axis).
- Physical characters of liposome would be like to list in new table or to insert into Table 3.
Response: The physical characters of liposomes were inserted into Table 3.
Sample name |
Area (ironed area) (µm2) |
Sa (µm) |
Sq (µm) |
Sp (µm) |
Sv (µm) |
Sy (µm) |
Roughness |
L1 |
910 |
0.070 |
0.090 |
0.338 |
-0.334 |
0.673 |
+ |
L2 |
912 |
0.099 |
0.126 |
0.473 |
-0.395 |
0.868 |
+ |
L3 |
929 |
0.234 |
0.312 |
1.193 |
-0.902 |
2.095 |
+++ |
L4 |
931 |
0.301 |
0.404 |
1.320 |
-1.034 |
2.354 |
+++ |
L5 |
917 |
0.101 |
0.126 |
0.394 |
-0.389 |
0.784 |
++ |
L6 |
905 |
0.087 |
0.108 |
0.368 |
-0.294 |
0.663 |
++ |
Where: Sa - Average Roughness, Sq - Mean Square Root Roughness, Sp - Maximum peak height, Sv - Maximum valley depth, Sy - Maximum peak-to valley height
- In flavonoid releasing experiment, several kinetic models have been analyzed. Please show detailed results or omit these descriptions.
Response: In order to investigate the mechanism of in vitro flavonoid release, several kinetic models have been analyzed (zero order, first order, Higuchi, Hickson-Crowell). The models were compared using (R2) correlation. The data showed in vitro flavonoid release kinetics in concordance with the Higuchi model, which approximated the experimental measurements well. The form of the Higuchi equation used for this study is the following:
(1)
Release kinetics modeling of liposomes loaded with Polygonum herba extract
Sample name |
R2 |
L3 |
0.9678 |
L4 |
0.9923 |
L5 |
0.9263 |
L6 |
0.8697 |
The resulting correlation quotient had a value of 0.9923 using the KinetDS 3 rev 2010 software.
Following are minor points:
- The company name “Schimadzu” in L146 and L343 is a mistake of “Shimadzu”.
Response; The correction was made.
- The section number “3.2” in L232 is a mistake of “2.3”.
Response: The correction was made.
- What is means “CDR(%)”? There was no abbreviation.
Response: CDR represents cumulative drug release.
- The cell density “104/cm2” is a mistake of “104/cm2”.
Response: The correction was made.
- In L304, two “microg” should be replace to “µg”.
Response: The correction was done.
- The antioxidant capacity of DPPH was explained to % value. How was the value calculated? And, what concentration was used of PAH in this assay?
Response: The inhibition ratio (%) was calculated from the equation:
% inhibition = [(Ac – As)/ Ac] x 100 (%),
where Ac –absorbance of control (DPPH with ethanol 50%), As – absorbance of PAH (0.1mg/mL) and the readings were made at 515 nm.
- “IC50” in L383 is a mistake of “IC50”.
Response: Done.
- Lipid name “PC” in L445 should be replaced into “phosphatidylcholine”.
Response: Done
- The section title” 3.6. The release of flavonoids entrapped in liposomes using a Franz cell system” makes it bold font.
Response: Done
- The releasing rate was showed in text (L527-529). What time were these values exhibited?
Response: The release rate was calculated for a period of 30 hours. The initial release rate is 70% in the first 6 hours after which the concentration of the pharmacologically active ingredient is maintained
- “in vitro” descriptions in L535, L537, L750, and L573 should be italic font.
Response: Done
- The word “polyphenols/ml” in L553 should be replaced into “GAE/ml”.
Response: Done

Round 2
Reviewer 1 Report
All the comments has been responded as per requirement. The manuscript can be accepted in its current form.